# PROMPT-GUIDED LOW-LEVEL RECOVERY AND HIGH-LEVEL FUSION FOR INCOMPLETE MULTIMODAL SENTIMENT ANALYSIS

## ABSTRACT

Multimodal Sentiment Analysis seeks to understand emotions by combining language, audio, and visual signals, but its real challenge lies in building models that stay robust when one or more modalities are missing or corrupted. Recent studies attempted to leverage available embedding to complement missing regions by single-level feature reconstruction or cross-modal fusion. However, both reconstruction-only and fusion-only pipelines are limited: the former amplifies noise from imperfect recovery, while the latter overlooks semantic restoration, leaving cross-modal gaps and complex intermodal relationships inadequately captured for robust generalization. To overcome these limitations, we propose Prompt-Guided Low-level recovery and High-level fusion (PGLH) for incomplete multimodal sentiment analysis, achieving deep cross-modal interactions from low-level semantic recovery to high-level semantic fusion through adaptive prompts. Specifically, PGLH mainly consists of Prompted Cross-Modal Masking (PCM2) and Unimodal-to-Bimodal Prompt Fusion (UBPF). First, PCM2 extends masked autoencoding to multimodal inputs by leveraging language-guided prompts to restore corrupted audio and visual tokens. This enables both structural fidelity and semantic grounding for low-level recovery. Secondly, in UBPF, self-guided prompts are introduced into each modality to extract fine-grained unimodal structures by selectively attending to informative regions. Next, they are progressively aligned with language-guided prompts for robust high-level fusion. Finally, PCM2 and UBPF realize the dual-level adaptation from low-level token reconstruction to high-level semantic integration, thereby effectively bridging modality gaps and more robust representations. Extensive experiments on MOSI, MOSEI, and SIMS demonstrate that PGLH consistently achieves impressive performance with missing data.

## 1 INTRODUCTION

Multimodal Sentiment Analysis (MSA) seeks to infer human emotions by jointly modeling language, audio, and visual signals. Compared with unimodal approaches, MSA leverages complementary cues that are essential for nuanced and reliable understanding in applications such as mental health monitoring, online education, and human–computer interaction. In real-world scenarios, however, multimodal streams are rarely complete: speech recordings are corrupted by background noise, faces are partially occluded, sensors fail intermittently, and transcripts contain errors. These imperfections create a significant challenge, as existing models often degrade sharply when inputs are incomplete or corrupted.

Prior studies have explored two main strategies to mitigate this problem. Reconstruction-based approaches (Yuan et al., 2021; Sun et al., 2023) attempt to restore missing features from available signals. For instance, EMT-DLFR (Sun et al., 2023) learns to complement incomplete inputs through low-level feature reconstruction. Fusion-based approaches (Li et al., 2024a;b; Zhang et al., 2024; Zhu et al., 2025) instead emphasize learning robust cross-modal representations directly. LNLN (Zhang et al., 2024) relies on the text modality as a dominant anchor, while P-RMF (Zhu et al., 2025) introduces proxy-driven latent modalities to account for uncertainty during integration. Both directions bring useful insights, yet each remains limited: reconstruction-only pipelines risk

amplifying noise from imperfect recovery, whereas fusion-only pipelines neglect the explicit restoration of corrupted semantics. As a result, large cross-modal gaps persist, and complex inter-modal relationships are only partially captured.

The impact of missing information can be observed at two distinct levels. At the low level, corrupted tokens such as discontinuous speech or blurred facial frames disrupt structural details, producing unreliable unimodal representations. At the high level, semantic discrepancies across modalities widen, as text, prosody, and facial expressions may misalign or conflict under high missing ratios. Without explicitly addressing both levels—restoring low-level semantics and aligning high-level semantics—multimodal fusion can easily overfit spurious patterns and fail to generalize.

To address these challenges, we propose Prompt-Guided Low-level recovery and High-level fusion (PGLH), a dual-level cross-modal adaptation framework that integrates token reconstruction with prompt-guided semantic fusion. PGLH consists of two complementary modules. The first, Prompted Cross-Modal Masking (PCM2), extends masked autoencoding to multimodal inputs and introduces language-guided prompts to reconstruct corrupted audio and visual tokens. By leveraging the semantic richness of text, PCM2 ensures structural fidelity and semantic grounding, providing reliable low-level recovery. The second, Unimodal-to-Bimodal Prompt Fusion (UBPF), focuses on high-level integration. It first employs self-guided prompts within each modality to emphasize informative and fine-grained cues, then progressively introduces language-guided prompts to align modalities with textual semantics, enabling coherent and discriminative fusion. Together, PCM2 and UBPF achieve dual-level adaptation, narrowing cross-modal gaps and producing representations that remain effective even when modalities are heavily incomplete.

Our main contributions are summarized as follows:

- We present PGLH, a unified framework for incomplete MSA that combines low-level recovery and high-level semantic alignment through prompt-based adaptation.

- We design PCM2, which extends masked autoencoding to multimodal data and leverages text-guided prompts for semantically faithful reconstruction of corrupted audio and visual features.

- We introduce UBPF, a progressive prompting strategy that refines unimodal structures and aligns them with textual semantics, enabling reliable cross-modal integration.

- Experiments on MOSI, MOSEI, and SIMS show that PGLH achieves impressive performance across both classification and regression tasks, while maintaining stability under varying degrees of missing data.

## 2 RELATED WORK

### 2.1 MULTIMODAL SENTIMENT ANALYSIS

Research on multimodal sentiment analysis (MSA) can be broadly divided into two lines: models designed under the assumption of complete modalities, and methods explicitly addressing corrupted or missing inputs. The first line assumes that all modalities are consistently available during training and inference (Zadeh et al., 2017; Tsai et al., 2019; Hazarika et al., 2020; Han et al., 2021; Yu et al., 2021; Liang et al., 2020). Such approaches mainly construct unified multimodal representations by modeling intra- and inter-modal interactions, using, for example, tensor-based fusion (Zadeh et al., 2017) or Transformer-based architectures (Tsai et al., 2019). While effective in controlled settings, their performance deteriorates sharply once one or more modalities are degraded or missing.

The second line of work targets incomplete or noisy modalities (Mittal et al., 2020; Yuan et al., 2021; Li et al., 2024a; Sun et al., 2023; Zhang et al., 2024; Zhu et al., 2025). Reconstruction-based strategies attempt to complement missing signals through feature completion or masked modeling (Yuan et al., 2021; Sun et al., 2023), but the imperfect quality of restored features often introduces additional noise. Fusion-based strategies (Li et al., 2024a;b), by contrast, bypass explicit imputation and instead focus on integrating the available modalities into a joint space. More recent designs emphasize semantic robustness: LNLN (Zhang et al., 2024) leverages the textual modality as a stable anchor, and P-RMF (Zhu et al., 2025) introduces proxy-driven latent modalities to handle uncertainty. These approaches improve resilience, yet most operate at a single level of abstraction—either

feature reconstruction or fusion—without jointly addressing both structural recovery and semantic alignment, which are especially critical under high missing ratios.

## 2.2 PROMPT LEARNING

Prompt learning has emerged as an effective paradigm for adapting large pre-trained models, initially in natural language processing where tasks are reformulated into prompt-based formats. Early work relied on handcrafted templates (Brown et al., 2020), while later methods such as prompt tuning (Lester et al., 2021) and prefix tuning (Li & Liang, 2021) introduced learnable prompts for flexible task adaptation. This paradigm has since been extended to vision and multimodal learning (Jia et al., 2022; Wang et al., 2022b; Khattak et al., 2023). For instance, MaPLe (Khattak et al., 2023) applies prompts across vision–language encoders to enhance alignment, and PromptFuse (Liang et al., 2022) develops modular prompt-based fusion for efficient multimodal integration.

Prompting has also been explored in the context of incomplete data. For example, Lee et al. (Lee et al., 2023) propose missing-aware prompts for visual recognition, showing their potential in handling absent modalities. Despite this progress, existing robust MSA methods such as LNLN (Zhang et al., 2024) and P-RMF (Zhu et al., 2025) still rely on text guidance or proxy-driven fusion, without explicitly employing prompts as adaptive intermediaries. In contrast, our framework PGLH integrates prompt learning into MSA for dual-level adaptation. The first module, Prompted Cross-Modal Masking (PCM2), extends masked autoencoding to multimodal data and incorporates text-guided prompts for semantically consistent token reconstruction. The second, Unimodal-to-Bimodal Prompt Fusion (UBPF), refines unimodal cues with self-guided prompts and progressively aligns them with language-guided prompts, achieving robust high-level fusion. Together, PCM2 and UBPF bridge the gap between structural recovery and semantic integration for incomplete multimodal sentiment analysis.

## 3 METHOD

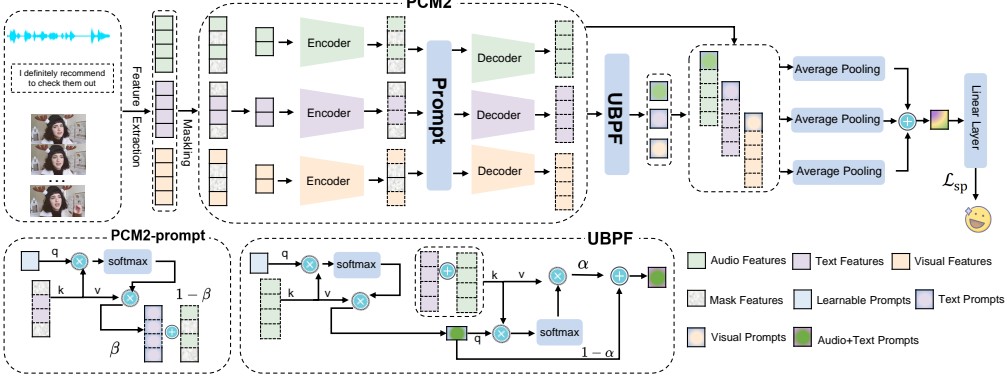

Figure 1: Overview of PGLH. Raw inputs are converted into token-level features and partially masked (0%–90%). Visible tokens are encoded by modality-specific Transformer encoders and projected into a shared latent space. During training, masked tokens are reconstructed with the help of text-guided prompting in the Prompted Cross-Modal Masking (PCM2) module, while the Unimodal-to-Bimodal Prompt Fusion (UBPF) module progressively refines unimodal features and aligns them with language-guided prompts for multimodal fusion.

## 3.1 FRAMEWORK OVERVIEW

We propose PGLH (Prompt-Guided Low-level recovery and High-level fusion), a dual-level cross-modal adaptation framework tailored for multimodal sentiment analysis under incomplete or noisy inputs. PGLH is composed of two complementary modules: Prompted Cross-Modal Masking (PCM2) for token-level recovery and Unimodal-to-Bimodal Prompt Fusion (UBPF) for adaptive semantic integration.

To simulate missing-modality conditions, random masking is applied to the input sequences with ratios sampled from $[0, 0.9]$. The remaining tokens, enriched with positional encodings, are encoded by modality-specific Transformer encoders (single-layer, pre-norm) and projected into a shared latent space. During training, PCM2 restores the masked positions by inserting learnable placeholders and decoding them with a lightweight Transformer decoder. Within this process, language-guided prompts provide semantic cues from text to support the reconstruction of audio and visual tokens, ensuring that the recovered features are both structurally consistent and semantically grounded. The reconstruction is supervised by two objectives: a global sequence reconstruction loss and a local masked reconstruction loss. At inference time, the decoder and prompt-guidance components are removed for efficiency, and the encoder outputs are directly forwarded to the subsequent module.

UBPF then refines and integrates the representations in two stages. First, self-guided prompts operate within each modality to emphasize informative regions and suppress noisy or redundant features, yielding clearer unimodal signals. Second, language-guided prompts are introduced to align these refined unimodal features with textual semantics, enabling robust high-level fusion across modalities. The resulting modality-level embeddings are aggregated by element-wise summation and passed into a regression head for sentiment prediction.

In summary, PCM2 focuses on low-level reconstruction guided by text semantics, while UBPF progressively adapts and fuses representations at a higher semantic level. Together, they achieve dual-level cross-modal adaptation, supporting accurate sentiment analysis under diverse missing rates while maintaining lightweight inference.

### 3.1.1 PROMPTED CROSS-MODAL MASKING (PCM2)

In multimodal sentiment analysis, acoustic and visual streams are particularly vulnerable to corruption caused by background noise, occlusion, or tracking errors. Directly training on such degraded inputs often leads models to memorize noise rather than capture meaningful patterns, while discarding incomplete modalities entirely results in loss of complementary cues that may still be informative. To address this dilemma, we design Prompted Cross-Modal Masking (PCM2), which extends the masked autoencoding paradigm to multimodal data and incorporates language-guided prompts into the reconstruction pipeline. PCM2 is intended to achieve two objectives simultaneously: (i) restore the structural continuity of corrupted sequences and (ii) enforce semantic grounding by exploiting text as a stable and sentiment-rich reference.

**Feature Extraction.** Each modality is first transformed into token-level features through well-established pretrained encoders: BERT Devlin et al. (2019) for textual input, Librosa McFee et al. (2015) for acoustic signals, and OpenFace Baltrušaitis et al. (2016) for visual frames. The feature sequence of modality $m \in \{t, a, v\}$ is denoted as:

$$Z_m \in \mathbb{R}^{T_m \times d^m}, \tag{1}$$

where $T_m$ represents the sequence length and $d^m$ the feature dimension. These embeddings preserve fine-grained temporal or spatial characteristics that form the basis for both structural recovery and semantic prompting.

**Masked Encoding.** To emulate incomplete or noisy conditions, random masking is applied using an indicator vector $\mathcal{M}_m \in \{0, 1\}^{T_m}$:

$$\tilde{Z}_m = \text{Mask}(Z_m, \mathcal{M}_m). \tag{2}$$

Only visible tokens are retained and fed into a modality-specific Transformer encoder:

$$H_m = \text{Encoder}(\tilde{Z}_m), \tag{3}$$

which outputs context-aware latent representations of the surviving subsequence. This step ensures that available signals are contextualized and organized before reconstruction is attempted.

**Structural Reconstruction.** The first stage of PCM2 focuses on recovering the corrupted sequences. Encoded features $H_m$ are concatenated with learnable mask tokens and decoded with a lightweight Transformer decoder:

$$\hat{Z}_m = \text{Decoder}\big(\text{Restore}(H_m, \mathcal{M}_m)\big), \tag{4}$$

where $\mathrm{Restore}(\cdot)$ reinserts placeholders at masked positions, ensuring the original sequence order is respected. This allows the model to predict missing tokens in a way that restores continuity and structural fidelity.

**Semantic Guidance with Language Prompts.** While structural recovery restores continuity, it does not by itself guarantee sentiment relevance. To inject task-specific semantics, PCM2 introduces language-guided prompts. For each non-text modality $m \in \{a, v\}$, a learnable prompt token $f_m^p$ attends to the textual representation $H_t$ through cross-attention:

$$Q_m = q(f_m^p), \quad K_m = k(H_t), \quad V_m = v(H_t), \tag{5}$$

$$\tilde{f}_m = \mathrm{softmax}\left(\frac{Q_m K_m^T}{\sqrt{d}}\right) V_m. \tag{6}$$

This operation extracts sentiment-rich cues from the text, which is typically the most reliable modality in multimodal sentiment analysis. The guided prompt is then reshaped and merged with modality features:

$$\tilde{H}_m = (1 - \beta) \cdot H_m + \beta \cdot \mathrm{Reshape}(\tilde{f}_m), \tag{7}$$

$$\hat{Z}_m = \mathrm{Decoder}(\mathrm{Restore}(\tilde{H}_m, \mathcal{M}_m)), \tag{8}$$

where $\beta$ is a learnable parameter that balances modality-specific content and textual guidance. By doing so, reconstruction is no longer a purely structural task but also a semantically grounded process.

**Training Objectives.** The reconstruction process is supervised by two complementary losses. The global reconstruction loss enforces consistency across the full sequence:

$$\mathcal{L}_{global} = \mathrm{MSE}(\hat{Z}_m, Z_m), \tag{9}$$

while the masked reconstruction loss emphasizes accuracy in corrupted regions:

$$\mathcal{L}_{mask} = \frac{1}{\sum \mathcal{M}_m} \sum_{i=1}^{T_m} \mathcal{M}_m^{(i)} \cdot \left\| \hat{Z}_m^{(i)} - Z_m^{(i)} \right\|_2^2. \tag{10}$$

Together, they enforce both local fidelity and global coherence, encouraging the model to recover sentiment-relevant details while maintaining the integrity of the sequence.

**Inference.** During inference, the decoder is omitted for efficiency. The encoder outputs, which have already been enriched with language-guided prompts during training, are directly passed to the Unimodal-to-Bimodal Prompt Fusion (UBPF) module. This design ensures that PCM2 not only restores incomplete modalities during training but also provides lightweight, semantically consistent representations that are well-suited for downstream multimodal fusion.

### 3.1.2 UNIMODAL-TO-BIMODAL PROMPT FUSION (UBPF)

Although PCM2 produces structurally consistent and semantically informed representations, these features are not yet fully aligned for sentiment prediction. To bridge this gap, we propose Unimodal-to-Bimodal Prompt Fusion (UBPF), a progressive refinement mechanism that preserves modality-specific information while promoting semantic coherence across modalities. UBPF operates in two stages: self-guided refinement and language-guided refinement, gradually transforming local unimodal cues into robust multimodal representations.

**Stage 1: Self-guided refinement.** Each modality $m \in \{t, a, v\}$ is initialized with a learnable prompt $f_m^{(0)}$, which interacts with its own reconstructed features $H_m$ through self-attention:

$$Q_m^{(0)} = q(f_m^{(0)}), \quad K_m^{(0)} = k(H_m), \quad V_m^{(0)} = v(H_m), \tag{11}$$

$$f_m^{(1)} = \mathrm{softmax}\left(\frac{Q_m^{(0)} K_m^{(0)T}}{\sqrt{d}}\right) V_m^{(0)}. \tag{12}$$

This stage reinforces intra-modal consistency by letting each prompt attend to the most informative regions within its modality. For example, an audio prompt may focus on pitch and intensity patterns, while a visual prompt may emphasize facial expressions. By retaining these discriminative cues, UBPF ensures that unimodal characteristics are not overshadowed before cross-modal alignment.

**Stage 2: Language-guided refinement.** While self-guided refinement secures modality-specific integrity, it does not guarantee semantic alignment. In the second stage, prompts are refined with text features as a semantic anchor. For text itself, refinement remains intra-modal:

$$f_t^{final} = \alpha_t \cdot \text{Attn}(f_t^{(1)}, H_t) + (1 - \alpha_t)f_t^{(1)}. \tag{13}$$

For audio and visual modalities, prompts are aligned jointly with their own features and the text representation:

$$f_m^{final} = \alpha_m \cdot \text{Attn}(f_m^{(1)}, [H_m; H_t]) + (1 - \alpha_m)f_m^{(1)}, \quad m \in \{a, v\}, \tag{14}$$

where $\alpha_t, \alpha_m$ is initialized to 0.5 and optimized during training. This step grounds acoustic and visual streams in textual semantics, ensuring that cross-modal fusion emphasizes sentiment-relevant patterns rather than noise or spurious correlations.

**Fusion and Prediction.** After refinement, final prompts are concatenated with their corresponding reconstructed features and aggregated into modality-level embeddings:

$$h_m = \text{AvgPool}([f_m^{final}; H_m]), \quad m \in \{t, a, v\}. \tag{15}$$

These embeddings are then combined via element-wise summation to form the final multimodal representation:

$$h = h_t + h_a + h_v, \qquad \hat{y} = \text{Linear}(h). \tag{16}$$

Through this process, UBPF preserves fine-grained unimodal structures while aligning them with textual guidance, yielding representations that are both discriminative and semantically coherent.

### 3.1.3 OVERALL LEARNING OBJECTIVE

The PGLH framework is optimized by jointly training reconstruction and prediction objectives:

$$\mathcal{L} = \lambda_{sp} \cdot \mathcal{L}_{sp} + \lambda_{mask} \cdot \mathcal{L}_{mask} + \lambda_{global} \cdot \mathcal{L}_{global}, \tag{17}$$

where $\mathcal{L}_{sp}$ is the sentiment prediction loss, $\mathcal{L}_{mask}$ the masked token reconstruction loss, and $\mathcal{L}_{global}$ the global sequence reconstruction loss. The coefficients $\lambda_{sp}$, $\lambda_{mask}$, and $\lambda_{global}$ are empirically set to 1, 0.05, and 0.05, respectively.

Here, $\mathcal{L}_{mask}$ improves recovery of corrupted regions, $\mathcal{L}_{global}$ preserves sequence-level coherence, and $\mathcal{L}_{sp}$ ensures discriminability for the target task. By integrating these objectives, PGLH achieves dual-level adaptation: low-level token reconstruction via PCM2 and high-level semantic refinement via UBPF. This combination enhances robustness under noisy or incomplete modalities and maintains efficiency at inference.

## 4 EXPERIMENTS

### 4.1 DATASETS AND EVALUATION SETTINGS

We evaluate PGLH on three standard multimodal sentiment analysis benchmarks: MOSI Zadeh et al. (2016), MOSEI Zadeh et al. (2018), and SIMS Yu et al. (2020), following prior work Zhang et al. (2024); Zhu et al. (2025). Each experiment is repeated with three random seeds, and results are reported using multiple metrics for comprehensive comparison.

MOSI contains 2,199 samples with language, audio, and visual modalities, split into 1,284 training, 229 validation, and 686 test instances, annotated on a $[-3, +3]$ sentiment scale. MOSEI includes 22,856 clips (16,326/1,871/4,659 for train/val/test) with the same scoring range. SIMS is a Chinese dataset of 2,281 clips from films and TV, split into 1,368/456/457, with sentiment labels in $[-1, +1]$. These datasets cover both English and Chinese, spontaneous and scripted scenarios, providing diverse challenges for multimodal sentiment analysis.

To evaluate robustness to incomplete inputs, we apply random masking with rates $r \in \{0.0, 0.1, \ldots, 0.9\}$, where a proportion $r$ of tokens in each modality is replaced with mask tokens during testing. Each configuration is repeated ten times, and averages are reported. Evaluation metrics include binary accuracy (Acc-2) and F1, mean absolute error (MAE), and correlation (Corr).

Table 1: Performance comparison on MOSI and MOSEI datasets. We report ACC-2, F1, ACC-5, ACC-7, MAE, and Corr.

| Method | MOSI | | | | | | MOSEI | | | | | |
| --- | --- | --- | --- | --- | --- | --- | --- | --- | --- | --- | --- | --- |
| | ACC-2 | F1 | ACC-5 | ACC-7 | MAE | Corr | ACC-2 | F1 | ACC-5 | ACC-7 | MAE | Corr |
| MISA | 70.33/71.49 | 70.00/71.28 | 33.08 | 29.85 | 1.085 | 0.524 | 75.82/71.27 | 68.73/63.85 | 39.39 | 40.84 | 0.780 | 0.503 |
| Self-MM | 69.26/70.51 | 67.54/66.60 | 34.67 | 29.55 | 1.070 | 0.512 | 77.42/73.89 | 72.31/68.92 | 45.38 | 44.70 | 0.695 | 0.498 |
| MMIM | 67.06/69.14 | 64.04/66.65 | 33.77 | 31.30 | 1.077 | 0.507 | 75.89/73.32 | 70.32/68.72 | 41.74 | 40.75 | 0.739 | 0.489 |
| CENET | 67.73/71.46 | 64.85/68.41 | 37.25 | 30.38 | 1.080 | 0.504 | 77.34/74.67 | 74.08/70.68 | 47.83 | 47.18 | 0.685 | 0.535 |
| TETFN | 67.68/69.76 | 63.29/65.69 | 34.34 | 30.30 | 1.087 | 0.507 | 67.68/69.76 | 63.29/65.69 | 47.70 | 30.30 | 1.087 | 0.508 |
| TFR-Net | 66.35/68.15 | 60.06/61.73 | 34.67 | 29.54 | 1.200 | 0.459 | 77.23/73.62 | 71.99/68.80 | 34.67 | 46.83 | 0.697 | 0.489 |
| ALMT | 68.39/70.40 | 71.80/72.57 | 33.42 | 30.30 | 1.083 | 0.498 | 77.54/76.64 | 78.03/77.14 | 41.64 | 40.92 | 0.674 | 0.481 |
| LNLN | 70.94/72.55 | 71.25/72.73 | 38.27 | 34.26 | 1.046 | 0.527 | 78.19/76.30 | 79.95/77.77 | 46.17 | 45.42 | 0.692 | 0.530 |
| P-RMF | 71.53/72.81 | 71.69/72.93 | 38.50 | 34.19 | 1.038 | 0.525 | 78.83/76.14 | 80.39/79.33 | 45.87 | 44.63 | 0.658 | 0.589 |
| **PGLH** | **71.90/73.33** | **72.84/73.32** | **39.34** | **34.34** | **1.033** | **0.533** | **78.85/78.52** | **80.85/79.94** | **47.68** | **46.76** | **0.653** | **0.593** |

Table 2: Performance comparison on SIMS dataset. We report ACC-2, F1, ACC-3, ACC-5, MAE, and Corr.

| Method | ACC-2 | F1 | ACC-3 | ACC-5 | MAE | Corr |
| --- | --- | --- | --- | --- | --- | --- |
| MISA | 72.71 | 66.30 | 56.87 | 31.53 | 0.539 | 0.348 |
| Self-MM | 72.81 | 68.43 | 56.75 | 32.28 | 0.508 | 0.376 |
| MMIM | 69.86 | 66.21 | 56.76 | 31.81 | 0.544 | 0.339 |
| CENET | 68.13 | 57.90 | 53.17 | 22.29 | 0.589 | 0.107 |
| TETFN | 73.58 | 68.67 | 56.91 | 33.42 | 0.505 | 0.387 |
| TFR-Net | 68.13 | 58.70 | 52.89 | 26.52 | 0.661 | 0.169 |
| ALMT | 71.85 | 76.21 | 56.47 | 34.16 | 0.509 | 0.372 |
| LNLN | 72.73 | **79.43** | 57.14 | 34.64 | 0.514 | 0.397 |
| P-RMF | 73.64 | 74.65 | 54.75 | 34.83 | 0.500 | 0.414 |
| **PGLH** | **74.36** | 74.84 | **57.23** | **35.45** | **0.496** | **0.423** |

MOSI and MOSEI further include multi-class accuracies (Acc-3, Acc-5, Acc-7), while SIMS reports Acc-3, Acc-5, and Corr, allowing assessment of both classification and regression performance across different granularity levels. All experiments are conducted on a 48G NVIDIA GPU with the 6000 Ada model. The model is trained for 200 epochs under random seeds 1111, 1112, and 1113, and the final results are averaged across these runs.

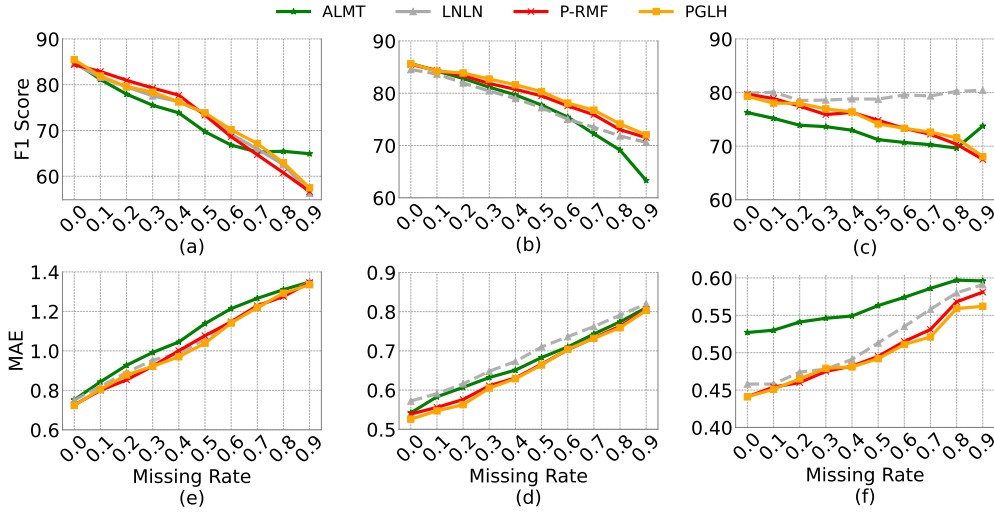

Figure 2: Performance of selected competitive models under different missing rates. (a)–(c) F1 scores on MOSI, MOSEI, and SIMS. (d)–(f) MAE on the same datasets. PGLH consistently shows superior robustness across varying modality incompleteness.

Table 3: Performance comparison on MOSI dataset. We report ACC-2, F1, ACC-5, ACC-7, MAE, and Corr.

| Method | ACC-2 | F1 | ACC-5 | ACC-7 | MAE | Corr |
|---|---|---|---|---|---|---|
| w/o PCM2-Decoder | 70.05/72.14 | 70.05/72.15 | 37.04 | 33.32 | 1.054 | 0.522 |
| w/o PCM2-Prompt | 71.47/72.83 | 71.4/72.86 | 38.08 | 33.46 | 1.088 | 0.516 |
| w/o UBPF | 69.3/71.23 | 69.3/71.33 | 37.20 | 32.19 | 1.059 | 0.515 |
| PGLH | **71.90/73.33** | **72.84/73.32** | **38.16** | **34.34** | **1.033** | **0.533** |

Table 4: Performance on MOSI dataset with $\mathcal{L}_{gobal}$ and $\mathcal{L}_{mask}$. We report ACC-2, F1, ACC-5, ACC-7, MAE, and Corr.

| $\mathcal{L}_{gobal}$ | $\mathcal{L}_{mask}$ | ACC-2 | F1 | ACC-5 | ACC-7 | MAE | Corr |
|---|---|---|---|---|---|---|---|
| | | 71.68/71.70 | 71.89/72.08 | 37.03 | 34.14 | 1.319 | 0.525 |
| ✓ | | 71.77/73.11 | 72.28/73.17 | 38.07 | 34.07 | 1.257 | 0.527 |
| | ✓ | 71.81/72.69 | 72.21/72.92 | 37.96 | 33.98 | 1.315 | 0.525 |
| ✓ | ✓ | **71.90/73.33** | **72.84/73.32** | **38.16** | **34.34** | **1.033** | **0.533** |

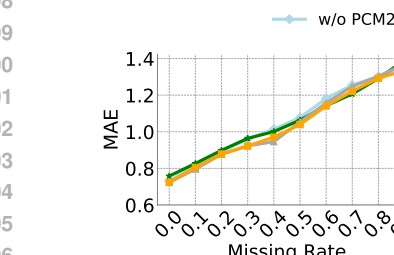 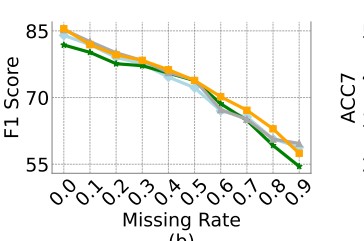 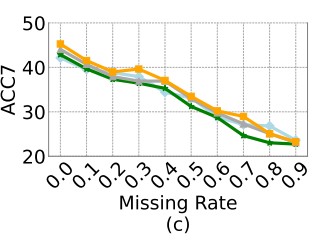

Figure 3: Ablation study of PGLH on MOSI under different missing rates. Curves show F1, MAE, and ACC-7 for the full model and variants without LEMPM, UBPF. Each component contributes to robustness under incomplete modalities.

## 4.2 ROBUSTNESS COMPARISON

We compare PGLH with a diverse set of competitive baselines, including MISA (Hazarika et al., 2020), Self-MM (Yu et al., 2021), MMIM (Han et al., 2021), CENET (Wang et al., 2022a), TETFN (Wang et al., 2023), TFR-Net (Yuan et al., 2021), ALMT (Zhang et al., 2023), LNLN (Zhang et al., 2024), and P-RMF (Zhu et al., 2025). Results on MOSI, MOSEI, and SIMS are reported in Tables 1 and 2.

On MOSI, earlier models such as MISA and Self-MM achieve acceptable binary accuracy but show clear weaknesses on fine-grained measures. More recent designs, including LNLN and P-RMF, bring notable gains, particularly in Acc-5 and Acc-7. PGLH pushes this trend further: it reduces MAE from 1.038 (P-RMF) to 1.033, raises correlation from 0.525 to 0.533, and achieves the best Acc-2 (73.33). This corresponds to a relative 1.5% improvement in correlation over P-RMF, suggesting that combining token-level recovery with prompt-guided fusion enhances both classification stability and regression accuracy.

On MOSEI, the contrast among baselines becomes sharper. ALMT delivers strong F1 and CENET achieves high correlation, but both fluctuate across other metrics. LNLN and P-RMF remain competitive, yet PGLH shows more balanced gains. It improves Acc-7 from 44.63 (P-RMF) to 46.76 and lowers MAE to 0.653, with the 4.8% relative gain on Acc-7 underscoring the benefit of uniting PCM2 for low-level semantic reconstruction with UBPF for high-level alignment.

On SIMS, spontaneous and noisy dialogues create a particularly demanding setting. LNLN reaches the best F1 (79.43), while P-RMF provides steadier performance across tasks. PGLH, however, achieves the strongest overall robustness: it secures the highest Acc-2 (74.36) and Acc-3 (57.23), while also lowering MAE to 0.496 and improving correlation to 0.423. Compared with P-RMF, the correlation gain is 0.9%, showing that PGLH is especially effective at leveraging textual prompts to guide cross-modal recovery in challenging conditions.

Figure 2 further illustrates performance under different missing ratios. Across all three datasets, PGLH consistently maintains lower MAE and higher F1 than the strongest baselines, confirming its adaptability to incomplete inputs.

In summary, PGLH does not only excel on a single metric but delivers consistent improvements across accuracy, correlation, and error measures. This balance highlights its ability to handle incomplete and noisy multimodal data in a unified way.

## 4.3 ABLATION STUDY

We further investigate the contribution of each component of PGLH through ablation experiments on MOSI. Figure 3 shows results under varying missing ratios, while Tables 3 and 4 present detailed comparisons. Removing the decoder from PCM2 leads to clear drops in performance: correlation falls from 0.533 to 0.522, F1 decreases, and MAE rises from 1.033 to 1.054. This indicates that the decoder is crucial for stabilizing token-level recovery when inputs are incomplete. Excluding the language-guided prompting within PCM2 also harms semantic restoration for audio and visual modalities, slightly increasing MAE (from 1.033 to 1.088) and lowering correlation (from 0.533 to 0.516), which underlines the value of text as guidance. Removing UBPF weakens cross-modal alignment, reducing F1 (from 73.32 to 71.33) and correlation (from 0.533 to 0.515), demonstrating that progressive two-stage refinement is essential for reliable fusion.

We also assess the role of reconstruction objectives. Eliminating both $\mathcal{L}_{global}$ and $\mathcal{L}_{mask}$ causes a sharp increase in MAE (from 1.033 to 1.319) and a large overall degradation, showing that these terms are critical for effective learning. Using only one of them provides partial gains, but combining both yields the strongest results, confirming their complementary effects: local recovery from $\mathcal{L}_{mask}$ and global consistency from $\mathcal{L}_{global}$.

Overall, these results highlight that each part of PCM2 with its decoder and language-guided prompts, UBPF with progressive refinement, and the dual reconstruction losses—plays an essential role. The ablation curves in Figure 3 further show that the complete model consistently achieves higher F1 and lower MAE across different missing ratios, confirming the benefit of combining low-level reconstruction with high-level prompting for incomplete multimodal sentiment analysis.

## 5 CONCLUSION

In this paper, we introduce PGLH, a dual-level cross modal adaptation framework designed to address the challenges posed by incomplete and noisy inputs in multimodal sentiment analysis. PGLH integrates both low level masked reconstruction and high level prompt guided fusion in order to produce stable and semantically consistent representations. The first module, Prompted Cross Modal Masking (PCM2), extends the idea of masked autoencoding by incorporating language informed prompts, which enables the recovery of corrupted acoustic and visual features while preserving both structural continuity and semantic grounding. Building on these reconstructed embeddings, the second module, Unimodal to Bimodal Prompt Fusion (UBPF), progressively refines unimodal features and aligns them with textual cues, leading to more adaptive and reliable multimodal integration. Extensive experiments on three widely used benchmarks, namely MOSI, MOSEI, and SIMS, demonstrate that PGLH consistently achieves strong performance across a range of missing ratios and noise conditions. These results confirm that unifying token level recovery with prompt guided cross modal adaptation not only improves local reconstruction but also strengthens global semantic fusion, resulting in representations that are both discriminative and robust. Looking ahead, we plan to explore more adaptive prompting mechanisms that can adjust to context and task dynamics, as well as extend PGLH to broader multimodal applications such as cross lingual sentiment analysis and real time interactive systems. These directions aim to further enhance the generalization ability and practical applicability of PGLH in diverse real world scenarios.

ETHICS STATEMENT

This work adheres to the ICLR Code of Ethics.[1] Our study does not involve human subjects, sensitive personal data, or potentially harmful applications. The proposed method does not introduce risks of misuse, privacy concerns, or discrimination.

REPRODUCIBILITY STATEMENT

We have taken extensive steps to ensure the reproducibility of our work. All datasets used in our experiments are publicly available and referenced in the paper. The implementation details are thoroughly described in the paper. We are committed to ensuring fairness and transparency in both experimentation and reporting. An code link of model training and testing will be released after paper acceptance to facilitate reproducibility of the reported results.

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

# A APPENDIX

## LLM USAGE STATEMENT

Large language models (LLMs) were used in this work as an assistive tool to improve the clarity and fluency of writing. All technical content, including model development and empirical evaluations, was conceived, implemented, and validated by the authors. The authors take full responsibility for the correctness and integrity of the paper.

## A.1 EVALUATION UNDER RANDOM MODALITY MISSING RATES

To assess the performance of PGLH under incomplete-modality conditions, we conduct experiments with random token masking ranging from 0% to 90% on MOSI, MOSEI, and SIMS datasets. Tables 5, 6, and 7 report comprehensive metrics including Acc-2, F1, multi-class accuracies (Acc-3, Acc-5, Acc-7), MAE, and correlation.

These results demonstrate that PGLH consistently maintains strong performance even as the proportion of masked tokens increases. The Cross-modal Prompt MAE (PCM2) ensures reliable low-level recovery of corrupted audio and visual features, while the Unimodal-to-Bimodal Prompt Fusion (UBPF) progressively refines unimodal cues and aligns them with textual guidance, achieving robust high-level fusion. Together, these modules preserve both local token-level structure and global multimodal semantic consistency, confirming the effectiveness of the dual-level prompt-guided adaptation strategy under challenging missing-modality scenarios.

Table 5: Performance of PGLH on MOSI under varying random missing rates.

| Missing Rate | ACC-2 | F1 | ACC-5 | ACC-7 | MAE | Corr |
|---|---|---|---|---|---|---|
| 0.0 | 82.07/85.69 | 82.14/85.47 | 50.87 | 45.26 | 0.7246 | 0.7949 |
| 0.1 | 80.76/82.28 | 81.03/81.91 | 47.52 | 41.53 | 0.8047 | 0.7369 |
| 0.2 | 79.15/79.94 | 79.70/79.61 | 46.79 | 38.98 | 0.8782 | 0.6750 |
| 0.3 | 78.28/78.52 | 79.07/78.36 | 44.9 | 39.62 | 0.9224 | 0.6293 |
| 0.4 | 76.68/76.33 | 77.71/76.25 | 43.15 | 37.06 | 0.9706 | 0.5951 |
| 0.5 | 73.18/73.84 | 74.42/73.85 | 38.48 | 33.48 | 1.0393 | 0.5394 |
| 0.6 | 67.20/70.12 | 68.73/70.18 | 35.13 | 30.21 | 1.1411 | 0.4519 |
| 0.7 | 64.29/66.57 | 66.01/67.14 | 30.47 | 28.95 | 1.2216 | 0.3700 |
| 0.8 | 60.93/62.71 | 62.70/62.96 | 30.76 | 25.08 | 1.2924 | 0.3099 |
| 0.9 | 56.41/57.32 | 56.91/57.47 | 25.36 | 23.25 | 1.3350 | 0.2278 |
| Average | 71.90/73.33 | 72.84/73.32 | 39.34 | 34.34 | 1.0330 | 0.5330 |

Table 6: Performance of PGLH on MOSEI under varying random missing rates.

| Missing Rate | ACC-2 | F1 | ACC-5 | ACC-7 | MAE | Corr |
|---|---|---|---|---|---|---|
| 0.0 | 83.67/85.01 | 83.67/85.66 | 53.72 | 51.31 | 0.526 | 0.771 |
| 0.1 | 82.84/84.61 | 82.69/84.21 | 52.77 | 50.73 | 0.547 | 0.755 |
| 0.2 | 82.36/83.31 | 82.49/83.87 | 51.44 | 49.55 | 0.563 | 0.732 |
| 0.3 | 81.09/82.46 | 81.71/82.71 | 48.84 | 48.33 | 0.605 | 0.686 |
| 0.4 | 80.17/80.42 | 81.27/81.61 | 48.86 | 47.32 | 0.629 | 0.657 |
| 0.5 | 78.45/78.38 | 81.01/80.31 | 46.81 | 46.42 | 0.664 | 0.606 |
| 0.6 | 77.13/76.55 | 80.36/78.11 | 45.95 | 45.39 | 0.704 | 0.544 |
| 0.7 | 75.71/74.29 | 79.66/76.76 | 43.97 | 44.85 | 0.732 | 0.482 |
| 0.8 | 74.38/71.34 | 77.97/74.12 | 42.25 | 42.77 | 0.759 | 0.401 |
| 0.9 | 72.67/68.81 | 77.67/72.04 | 42.19 | 40.97 | 0.803 | 0.291 |
| Average | 78.85/78.52 | 80.85/79.94 | 47.68 | 46.76 | 0.653 | 0.593 |

Table 7: Performance of PGLH on SIMS under varying random missing rates.

| Missing Rate | ACC-2 | F1 | ACC-3 | ACC-5 | MAE | Corr |
|---|---|---|---|---|---|---|
| 0.0 | 78.42 | 79.29 | 66.88 | 40.12 | 0.441 | 0.547 |
| 0.1 | 77.24 | 78.03 | 63.51 | 40.92 | 0.451 | 0.532 |
| 0.2 | 76.67 | 77.99 | 62.56 | 38.73 | 0.465 | 0.516 |
| 0.3 | 75.97 | 76.95 | 60.81 | 38.00 | 0.479 | 0.494 |
| 0.4 | 75.75 | 76.39 | 60.67 | 35.67 | 0.481 | 0.471 |
| 0.5 | 74.92 | 74.15 | 57.46 | 34.43 | 0.492 | 0.448 |
| 0.6 | 73.44 | 73.33 | 54.61 | 33.95 | 0.511 | 0.403 |
| 0.7 | 72.78 | 72.62 | 50.16 | 32.39 | 0.521 | 0.371 |
| 0.8 | 70.56 | 71.55 | 47.98 | 30.46 | 0.559 | 0.287 |
| 0.9 | 67.84 | 68.05 | 47.68 | 29.85 | 0.562 | 0.159 |
| Average | 74.36 | 74.84 | 57.23 | 35.45 | 0.496 | 0.423 |

## A.2 EXPLORATION OF LOSS WEIGHTS

We evaluate the effect of different combinations of masked reconstruction loss ($\mathcal{L}_{mask}$) and global reconstruction loss ($\mathcal{L}_{global}$) on MOSI. The setting $\lambda_{mask} = 0.05$ and $\lambda_{global} = 0.05$ achieves the most balanced performance, and we adopt the same weights for all datasets.

Table 8: Impact of different loss weight combinations on MOSI.

| $\lambda_{mask}$ | $\lambda_{global}$ | ACC-2 | F1 | ACC-5 | ACC-7 | MAE | Corr |
|---|---|---|---|---|---|---|---|
| 0.025 | 0.025 | 71.54/71.53 | 70.62/71.59 | 38.83 | 34.18 | 1.041 | 0.535 |
| 0.025 | 0.05 | 71.85/72.77 | 71.94/72.76 | 38.03 | 34.29 | 1.036 | 0.514 |
| 0.05 | 0.05 | 71.90/73.33 | 72.84/73.32 | 38.16 | 34.34 | 1.033 | 0.533 |
| 0.05 | 0.1 | 72.07/73.29 | 71.96/73.27 | 37.74 | 34.34 | 1.037 | 0.523 |
| 0.1 | 0.1 | 71.98/73.02 | 71.83/72.99 | 38.06 | 34.24 | 1.038 | 0.523 |

