# OpenReview forum: "Prompt-Guided Low-Level Recovery and High-Level Fusion for Incomplete Multimodal Sentiment Analysis"
_ICLR.cc/2026/Conference — ICLR 2026 Conference Withdrawn Submission_

### Official Review · Reviewer_MY9E · 2025-10-20

**Soundness:** 1
**Presentation:** 1
**Contribution:** 1
**Rating:** 2
**Confidence:** 5

**Summary:**

This paper proposes PGLH, a framework designed for Multimodal Sentiment Analysis (MSA) with incomplete data. The method operates at two levels: a low-level module (PCM2) that uses language-guided prompts to reconstruct missing tokens in audio and visual streams, and a high-level fusion module (UBPF) that uses prompts to refine and align multimodal representations for sentiment prediction. While the problem is relevant, the paper suffers from significant flaws in its positioning, methodology, and experimental evaluation.

**Strengths:**

1. The method is easy to understand.

2. The problem for missing data of MSA is important.

**Weaknesses:**

1. Originality and Novelty Concerns: The paper's most critical flaw is its failure to acknowledge or discuss highly relevant prior work. The field of prompt-based learning for incomplete MSA has been explored previously, notably in works like "Multimodal Prompt Learning with Missing Modalities for Sentiment Analysis and Emotion Recognition" (ACL 2024) and subsequent research. This paper completely overlooks this entire line of prompt-based works for incomplete MSA. As a result, the claims of novelty are unsubstantiated, and the proposed method appears similar to existing approaches without a clear differentiation. This omission raises concerns about the thoroughness of the literature review and the actual contribution of this work.

2. Questionable Experimental Setup: The choice to evaluate the model on "token missing" instead of the more commonly studied "modality missing" is poorly justified. In real-world applications, it is far more common for an entire modality (e.g., the audio track or visual feed) to be unavailable than for sporadic, random tokens within a stream to be missing. This experimental design choice creates a disconnect from the problem as it is typically framed in the literature. Furthermore, it calls into question the fairness of comparing PGLH against baselines that were designed to handle modality-level absence. The paper does not provide a compelling reason for this deviation.

3. Insufficient Experimental Depth: The experiments presented are superficial and lack the depth required to be convincing. A central claim of the paper is the effectiveness of the low-level reconstruction via the PCM2 module. However, the authors provide no qualitative evidence to support this. Visualizations or qualitative examples of the reconstructed audio or visual data are essential to validate that the module is performing meaningful recovery rather than simply learning to output generic features. The evaluation is limited to reporting final metrics and a basic ablation study, which is not sufficient to understand the model's behavior or substantiate its claimed mechanisms.

**Questions:**

See weaknesses.

---

### Official Review · Reviewer_TqCy · 2025-10-23

**Soundness:** 2
**Presentation:** 3
**Contribution:** 2
**Rating:** 2
**Confidence:** 4

**Summary:**

This paper focuses on incomplete multimodal sentiment analysis and proposes PGLH, which leverages PCM2 for low-level recovery and UBPF for high-level integration. The reported results on three datasets demonstrate the effectiveness of the proposed approach.

**Strengths:**

1.	The paper addresses an important and meaningful task: incomplete multimodal sentiment analysis.

2.	Overall, the paper is well-structured and clearly written.

**Weaknesses:**

1.	The paper does not provide compelling motivations for its contributions.

a)	Firstly, it claims that the proposed method can “enable both structural fidelity and semantic grounding for low-level recovery”. However, there is a lack of experimental comparisons with existing methods in terms of reconstruction performance.

b)	Secondly, the paper argues that “reconstruction-only pipelines risk amplifying noise from imperfect recovery, whereas fusion-only pipelines neglect the explicit restoration of corrupted semantics.”Previous reconstruction-based approaches have also made significant efforts to utilize multimodal information for accurate reconstruction, and the proposed prompt-based method is, in essence, another form of multimodal fusion. Therefore, the paper should further clarify why and how the proposed approach better addresses the challenges posed by imperfect recovery.

2.	The paper simulates missing modalities through random masking. It remains unclear how this artificial setup correlates with real-world scenarios of missing or noisy data. Additional experiments using more realistic missing or noisy conditions are necessary to validate the practical applicability of the method.

3.	The experimental section lacks sufficient evaluations to convincingly demonstrate the effectiveness of the proposed methods. More baseline methods should be included for comprehensive comparison.

4.	The performance improvements reported in Table 1 and Table 2 over SOTA methods are relatively marginal. Statistical significance tests should be conducted to assess whether these gains are remarkable.

5.	From my perspective, the paper primarily introduces a prompt-based approach to incomplete multimodal sentiment analysis. However, previous studies have already explored a variety of multimodal fusion techniques, and the proposed prompt-based method can also be viewed as falling within the broader category of multimodal fusion. However, from my perspective, multimodal fusion is undoubtedly important but not a promising route to address incomplete multimodal sentiment analysis. This is further supported by the marginal performance gains observed in the experiments, which do not strongly justify the novelty or practical advantages of the proposed method.

**Questions:**

See weakness

---

### Official Review · Reviewer_CoFN · 2025-10-29

**Soundness:** 2
**Presentation:** 2
**Contribution:** 2
**Rating:** 2
**Confidence:** 4

**Summary:**

This paper presents PGLH, a dual-level cross-modal framework that handles incomplete and noisy inputs in multimodal sentiment analysis. By combining low-level masked reconstruction with high-level prompt-guided fusion, PGLH achieves stable and semantically consistent representations. Experiments on MOSI, MOSEI, and SIMS show consistent improvements over baselines across different missing rates.

**Strengths:**

1. The coexistence of complete and incomplete modalities is a common issue in real-world multimodal applications, making the research problem both practical and valuable.
2. The experiments are comprehensive, covering three datasets and various missing rates with detailed comparisons, ablations, and hyperparameter studies, providing empirical support for the method.

**Weaknesses:**

1. The paper does not sufficiently discuss implementation details and sensitivity analysis regarding prompt vector design, including its dimension, initialization, number, and positional placement (e.g., before, within, or after the token sequence).
2. It lacks an explicit explanation and supporting experiments on why the decoder can be safely removed during inference (e.g., whether the encoder has implicitly learned reconstruction ability). A comparison with partially retaining a lightweight decoder would help justify this design choice.
3. At line 222, the definition of f_m^p is unclear and should be explicitly specified.

**Questions:**

1. At line 298, the sentiment prediction loss is introduced but not explained earlier in the text, its exact application point and functional role are not described.
2. The paper lacks theoretical or interpretive justification for why the prompt can effectively guide reconstruction under varying missing rates and across different modalities. Visualization analyses (e.g., attention maps or reconstruction examples) would improve interpretability.
3. In Table 4, the authors omit an ablation analysis of the sentiment prediction loss.

---

### Official Review · Reviewer_xx2U · 2025-10-31

**Soundness:** 3
**Presentation:** 3
**Contribution:** 3
**Rating:** 6
**Confidence:** 5

**Summary:**

This paper proposes a framework PGLH to address the key challenges in incomplete MSA. It uses "prompts" to achieve dual-level cross-modal adaptation. It first recovers corrupted audio and visual features using language-guided prompts for low-level reconstruction, then performs high-level fusion by progressively aligning modalities through self-guided and language-guided prompt fusion. Experiments on the MOSI, MOSEI, and SIMS datasets demonstrate that PGLH achieves robust and superior performance under various missing rates.

**Strengths:**

1. Decomposes the problem into "low-level" and "high-level", and use "prompt" as a unified mechanism that through the entire pipeline from low-level to high-level.
2. The designs of PCM2 and UBPF are reasonable, and experiments have also proven their effectiveness.
3. Good performance.

**Weaknesses:**

1. Some descriptions are unclear. For example, the definition and role of "prompt" in the Abstract and Introduction are not clearly explained. It may lead readers to mistake it for prompts used in LLMs. After reading the "Method" section, I was able to understand the specific application of the term "prompt" in this paper.
2. Lacks visualization: (1) Is there any way to verify that the prompt representations are useful as intended? (2) Could you show the intermediate attention maps and analyze them? (3) Could you show some representative cases where the model's predictions succeeded or failed?
3. There are problems with the citation. For example, Zadeh et al., 2017 and Han et al., 2021 (line 95), Zhang et al., 2023 (line 418), have been published but are still cited using arXiv versions. Additionally, ALMT is incorrectly cited as LNLN on lines 51-52 and line 101.
4. Could you clarify in the caption of Figure 1 that PCM2 and UBPF are shown with audio modality streams as an example?
5. What would happen if the language-guided prompts in PCM2 were replaced with vision-guided or audio-guided prompts?

**Questions:**

Please see Weaknesses.

---

### Note · Authors · 2025-11-13

I have read and agree with the venue's withdrawal policy on behalf of myself and my co-authors.